# Contact-Free Support Structures for the Direct Metal Laser Melting Process

**DOI:** 10.3390/ma15113765

**Published:** 2022-05-25

**Authors:** Alican Çelik, Emre Tekoğlu, Evren Yasa, Mehmet Şeref Sönmez

**Affiliations:** 1General Electric Aviation, Gebze 41400, Turkey; emre.tekoglu@ge.com; 2Department of Metallurgical and Materials Engineering, Faculty of Chemical and Metallurgical Engineering, Istanbul Technical University, Istanbul 34467, Turkey; ssonmez@itu.edu.tr; 3Department of Mechanical Engineering, Eskisehir Osmangazi University, Eskisehir 26040, Turkey; eyasa@ogu.edu.tr

**Keywords:** contact-free supports, overhang, L-PBF, direct metal laser melting, roughness, Co-Cr-Mo alloy

## Abstract

Although Direct Metal Laser Melting (DMLM), a powder bed fusion (PBF) Additive Manufacturing (AM) for metallic materials, provides many advantages over conventional manufacturing such as almost unlimited design freedom, one of its main limitations is the need for support structures beneath overhang surfaces. Support structures are generally in contact with overhang surfaces to physically prop them up; therefore, they need to be removed after manufacturing due to not constituting a part of the main component design. The removal of supports is a process sequence adding extra time and cost to the overall manufacturing process and could result in damaging the main component. In this study, to examine the feasibility of contact-free supports for overhang surfaces in the DMLM process, coupons with these novel types of supports were prepared from CoCrMo alloy powder. This study aims to understand the effect of two parameters: the gap distance between supports and overhang surfaces and the inclination angle of overhang surfaces, on the surface topography and microstructural properties of these surfaces. Visual inspection, roughness measurements, and optical microscopy were utilized as characterization methods The roughness parameters (Ra, Rq, and Rz) were obtained using the focus variation method, and optical microscope analysis was performed on the cross-sections of the overhang surfaces to investigate the sub-surface microstructure and surface topology. Results showed that contact-free supports have a positive effect on decreasing surface roughness at all build angles when the gap distance is correctly set to avoid sintering of the powder in between the overhang and supports or to avoid too large gaps eliminating the desired effect of the higher thermal conductivity.

## 1. Introduction

Powder Bed Fusion (PBF) is one of the Additive Manufacturing (AM) technologies that enables parts are difficult to make by conventional manufacturing routes due to its very complex geometries in conjunction with it can be produced from a wide range of materials [1,2]. PBF technologies including Electron Beam Melting (EBM) and Direct Metal Laser Melting (DMLM) provide many advantages such as weight reduction, low buy-to-fly ratios, a high level of customization, simplified supply chains, a reduced need for joining and assemblies, etc. in comparison to conventional manufacturing routes. Moreover, PBF technologies lead to higher dimensional accuracy and a feature resolution among other AM processes [3,4,5,6,7]. The main principle of PBF is to produce a final model, layer by layer, in a powder bed by using the thermal energy of a heat source. Principally, there are two types of thermal energy: laser beam and electron beam. Direct Metal Laser Melting (DMLM) constitutes the powder bed fusion modality with Selective Laser Sintering (SLS) or Electron Beam Melting (EBM) technologies. Although DMLM, DMLS (Direct Metal Laser Sintering), and SLM (Selective Laser Melting) have the same working principle of utilizing a laser beam as a heat source during the process, they are denominated distinctly by different tradenames. The DMLM process starts with CAD data preparation of the part to be built, then Standard Tessellation Language (STL) data is obtained from this CAD data. In the next step, the STL file is sliced in the desired layer thickness and a laser path is generated for every single layer. The generated slice data is transferred to the DMLM machine, then the first step of building coats a designated layer thickness of powder on the build plate which is fastened to the build chamber of the machine. Subsequently, a laser with a specified power level set in the parameter starts to operate by melting the metal powder selectively along the path in the corresponding slice layer by hitting on the prepared powder bed. Later, melted paths solidify and fuse. Once the laser finalizes the scanning of one layer, the build plate is lowered by one layer thickness and the powder hopper is elevated more than the build plate. The elevated value is defined by multiplying the layer thickness with the dosing factor. Through this elevation difference between the build plate and the powder hopper, powder loss or shrinkages are compensated. The same process sequence takes place and a new layer is built up on the previous melted-solidified one. The process of the building up of layers is repeated until all parts are wholly produced [8].

One of the major problems in PBF processes, however, is the relatively high surface roughness values encountered, especially on overhang surfaces [9,10,11]. As is well known, surface roughness is one of the important surface conditions determining the fatigue performance of engineering materials [12,13]. In other words, high surface roughness will feature height peaks and deep valleys, which are potential stress concentration zones resulting in crack formation and consequent propagation [14,15]. Therefore, it is important to reduce the surface roughness of the parts manufactured by PBF, especially for applications in need of long fatigue life. Generally, surface treatments such as chemical milling [16,17], CNC machining [18,19], and abrasive flow machining [20,21] are applied as a post-processing step, after the PBF process, to overcome this surface quality problem. However, the complexity of the PBF parts, which is one of the most important strengths of AM, generally inhibits uniform material removal throughout the whole process or leads to a line-of-sight problem at some features like internal cooling channels [22]. Thus, there remains an intensive effort to substantiate a surface treatment method that enables homogenous material removal with an effective line-of-sight. Since leveraging a material removal method for complex PBF parts continues, it is crucial to produce those parts with low surface roughness as much as possible without any support structures. Further studies are required within the context of process parameter optimization such as lowering the laser power or increasing the scan speed, to decrease the surface roughness [23,24]. Moreover, other precautions such as a contour laser scan following the core (hatch/inskin) laser scan provide a smoother surface by re-melting the rougher pre-solidified surface during the core laser scan [25,26,27]. During PBF, the surface roughness is determined by the predominance of different mechanisms such as stair stepping [28], balling effect [29], powder sticking [27], etc. at different build orientations. As implied, the overhang surfaces of the PBF parts exhibit severe surface roughness because overhang areas are free-standing surfaces, and these layers are scanned over the powder bed rather than a solidified layer beneath or the base plate [30]. The overhang problem is more severe with the L-PBF process, which is also known as Direct Metal Laser Melting (DMLM) due to relatively lower preheating temperatures and loose powder bed in comparison to the Electron-PBF process where very high preheating temperatures are used and the loose powder around the part to be produced is sintered to avoid smoke formation [31]. Several defects such as warping, dross formation, and staircase effect are formed on the overhang surface under a high heating/cooling cycle of L-PBF, and overhang surfaces end up with a poor surface finish [32]. Thus, the overhang surfaces having a critical angle need to be supported by structures that are removed after the process leaving connection marks and deteriorating the surface characteristics. The support structures used for overhang surfaces are a waste of time, material, and post-processing efforts. It is clear that the above-mentioned surface treatment methods to remove the supports add additional problems such as a poor surface finish at overhang regions because of “macroscale” support remnants [33,34]. Therefore, there are many studies focused on understanding the behavior of support structures and their effects on the main part surface, and how to minimize their existence in the design stage [33,34,35,36,37,38,39,40,41].

There is a novel type of support structures called “contact-free” or “contactless” supports. These supports do not touch the main part’s overhang surface but still provide the necessary heat evacuation to some extent during printing [42]. Given that the relevant support structures are not in contact with the main part, the overhang surfaces could be produced with a better surface finish. Moreover, time and cost arising from the subsequent support removal post-processes could be reduced [42]. In addition, at some locations, depending on the complexity level of the part geometry, it may be impossible to remove the supports. Xiang et al. [43] obtained results indicating that melt pool properties are affected considerably by the oblique angle which is inferred as the angle between the x-axis of the build plate and the overhang feature of the part. It is found that the melt pool length along the scan direction is longer on the overhang feature because of the lower thermal conductivity of the powder bed in comparison to the bulk material [43]. In light of this aspect, Cooper et al. examined the outcomes of contact-free support structures by replacing them with conventional support structures making contact with overhangs in the E-PBF process and attained reasonable improvement in terms of distortion of overhangs [42]. Paggi et al. [44] studied contact-free supports for Ti Gr23 powder in the L-PBF process aiming at 0, 30, and 45 degrees of overhang angles. The outcomes indicated that using contact-free supports significantly reduced the warping but the average roughness did not improve. As shown here, there is a very limited number of studies on contact-free supports although they are very promising to remove the need for post-processing and to widen the application area of L-PBF.

In this study, in order to further elaborate on the impact of contact-free supports, the effect of the contact-free support’s design features on the overhang surface characteristics was investigated for Co-Cr-Mo powder in the L-PBF process. Contactless support structures are located beneath the overhang surfaces having different inclination angles of 15°, 25°, 30°, and 45° with different gap distances of 100 μm, 125 μm, 150 µm, 250 µm, and 300 µm. Surface characteristics of the overhang surfaces printed with different support characteristics are identified through roughness measurements and microstructural characterizations.

## 2. Materials and Methods

In this study, gas atomized spherical Co-Cr-Mo powder (Praxair^TM^ CO-538, Indianapolis, IN, USA) having an average particle size of 33 µm was used as the raw material. The chemical composition of the used powder is shown in Table 1 and it conforms to ASTM F3213-17 standard. 

CoCrMo alloys are characterized by impressive toughness and strength, corrosion, and wear resistance [45,46,47]. Due to these characteristics, these alloys are utilized for applications at elevated temperatures and specific strengths such as wind turbines and jet-engine components. Moreover, they are widely used for various orthopedic implants such as bone-fixation tools, entire hip- and knee-joint replacements along with dental applications such as dental prostheses and restorations, e.g., dental crowns [48,49,50,51]. Due to their specific characteristics and wide application range, CoCrMo alloys are the more utilized alloys in PBF. Therefore, this material is selected for this study in order to see the effects of this novel type of support on this alloy.

Samples were designed in NX 12.0.2 to understand the effect of different oblique angles and gap distances on the surface roughness. Figure 1 shows the representation of design samples where the black arrow indicates the powder coating direction. In the build lay-out, overhang surfaces with 15°, 25°, 30°, and 45° oblique angles are printed whereas the gap distances concur as 100 µm, 125 µm, 150 µm, 250 µm, and 300 µm for each oblique angle. The gap distances are selected as folds of the layer thickness which is 50 µm. In addition, unsupported samples are added to the build plan for comparison. The sample codes are listed in Table 2 and each sample number represents a unique combination of gap distance and oblique angle.

A Concept Laser^TM^ M2 Cusing (Lichtenfels, Germany) machine was utilized to produce the modeled geometries under a protective atmosphere of nitrogen. In addition, 180 W laser power, 1500 mm/s laser speed, and 60 µm hatch spacing parameters were used with the layer thickness set to 50 µm [52].

After printing, SFM-AT800 system (Solukon^TM^, Augsburg, Germany) was used to evacuate the remaining trapped powder particles from the built specimens. Afterward, wire electrical discharge machining (WEDM) is employed via AgieCharmill Cut 20 P WEDM (GF+^TM^, Biel, Switzerland) system to separate the specimens from the build plate.

The surface roughness of the produced parts was evaluated by an Alicona^TM^ InfiniteFocus (IF) G5 instrument relying on focus variation technique. During the measurements, the magnification was set to X5 while the illumination type was selected as polarized coaxial. The lateral and vertical resolution values were determined as 6 µm and 900 nm, respectively. Before the measurements of surface roughness parameters (Ra, Rq, and Rz), the 3rd degree polynomial form removal was performed on 3D-view dataset. Finally, 2D surface profile lines with a cut-off length of 800 µm were extracted from the form removed dataset to calculate roughness parameters (Ra, Rq, and Rz). 3 different profile lines were extracted for each sample and surface parameter and the calculated Ra, Rq, and Rz are reported with their standard deviations. In addition, each scanned surface was imported into GOM Inspect 2019 in STL extension with their corresponding CAD geometries. Afterward, the nominal and real scan geometries were aligned by 3-2-1 alignment method to see the deviation of the scanned surface profile from nominal surface based on surface texture map. Metallographic studies were performed on cross-sections of the parts to directly observe the surface profile of the overhang surfaces. Samples were cut up via a Struers^TM^ Secotom (Copenhagen, Denmark) cutter from its middle location. After cutting, all samples were mounted in a Struers^TM^ CitoPress mounting machine. Struers^TM^ Tegramin was used for polishing. Cross-sectional micrographs were obtained via an optical microscope, namely Nikon^TM^ Eclipse MA200 (Tokyo, Japan). Finally, surface topographies of the samples were examined by Zeiss MERLIN FE—SEM (field emission—scanning electron microscope) under 15 kV acceleration voltage.

## 3. Results

Figure 2 shows the contact-free supported samples manufactured by the L-PBF method. It is clearly seen from Figure 2 that some of the contact-free structures are fused to the overhang surfaces and could not be manually removed from the specimens. Therefore, the parts which could not be separated from the support structures or those separated but still exhibited a very poor surface quality were not involved in the characterization studies. Regarding the oblique angle of 45° of overhang surfaces, contact-free support structures were stuck on overhang surfaces at 100 and 125 µm of gap distances. In addition to 100 and 125 µm of gap distances, contact-free support structures were also stuck on the overhang surface of 30° at 150 µm of gap distance. When the oblique angle was decreased to 15°, all samples were unsuccessful. For a gap distance of 100 µm, the support structure and the main part were fused and for the rest of the gap distances, the surface quality of the overhang surfaces significantly deteriorated. It is seen that the fusion phenomenon is observed at relatively low gap distances and/or build angles. It is assumed that when the build angle gets lower at the overhang’s outermost layer, the freestanding melt pool deeply penetrates the powder bed. In addition to this reason, it is well known that the heat accumulation per area increases with decreasing build angle, which also serves as a catalyzer for the fusion of the support structure and overhang. On one hand, it could be stated that decreasing the gap distance between the contactless support and overhang surface increases the heat dissipation and thereby, decreases the possibility of fusion between support and overhang surface. Yet, the fusion taking place at lower gap distances in this study proves that the contact-free support does not suffice to maintain unsticking. Therefore, it could be stated that the dramatic intrusion of the melt pool dominates the heat dissipation provided by contact-free structures at lower gap distances. Finally, as mentioned above, when the build angle decreases from 45° to 30°, contact-free structures could not tolerate sticking problems at even 150 µm of gap distance. It is also important to mention that another set of coupons was printed by the L-PBF method in the same print having an oblique angle of 25°, which is between 15° and 30° at 250 and 300 µm gap distances. In other words, 3 different sets of coupons, which are identified as US-25°, CS-25°-250 µm, and CS-25°-300 µm were successfully printed and successfully separated from the contact-free supports as shown in Figure 2 as well.

Table 3 lists the Ra, Rq, and Rz values obtained from the overhang surfaces for different oblique angles (15°, 30° and 45°) and gap distances (0, 100, 125, 150, 250, and 300 µm). The overhang surfaces, which were fused to the support or exhibited poor surface quality are pointed out in Table 3 as well. As expected, Ra, Rz, and Rq results of the overhang surfaces tend to increase with decreasing oblique angle. Regardless of the overhang surfaces being unsupported or supported, it is seen that Ra decreases by about 200–250% when the oblique angle decreases from 45° to 30°. Yet, when the oblique angle decreases from 45° to 30°, it is observed that the deterioration of the Ra is slightly higher (~50%) at the unsupported overhang surfaces compared to those of contact-free supported ones. This shows that the contact-free support structures prevent the rate of deterioration of the surface quality to some extent by decreasing the oblique angle. On the other hand, a change in the gap distance also influences the surface roughness values. Independent of the oblique angle, Ra of the overhang surfaces decreases at 250 and 300 µm gap distances. In other words, Ra roughness of the CS-45°-250 µm and CS-45°-300 µm is 15% lower than those of unsupported ones. However, Ra of the CS-45°-150 µm is almost identical to US-45° showing that the effective gap distance is close to 250 and 300 µm. When it comes to the CS-30°-250 µm and CS-30°-300 µm, Ra decreases by 23% and 39%, respectively, compared to the unsupported overhang surfaces. As also seen in Table 3, Ra of the US-25° is 67 and 93% higher than those of CS-25°-250 µm and CS-25°-300 µm, respectively, and the decreasing trend in surface roughness by the implementation of contact-free support structures at 250 and 300 µm gap distances is more remarkable compared to the Ra of 30° and 45°.

Figure 3a–d shows different cross-sectional OM images of unsupported overhang surfaces. As could be seen from Figure 3a, a protrusion, which is squared as green, is observed at the overhang surface of US-45°. The height of the melt pool is approximately 150–200 µm and a fused particle is observed at the tip of the melt pool (Figure 3b). It is considered that the relevant liquid melt pool exists at a free standing layer protruded into the powder bed opposite to the build direction and a fusion occurred between the tip of the melt pool and particles existing in the powder bed during solidification. In other words, the larger heat accumulation at lower oblique angles favors the deep penetration of the melt pool into the powder below based on high capillary forces, and the overhang surface ends up with a dross formation, which is an indication of poor surface quality as also observed by other researchers [53]. Regarding the cross-sectional OM images of US-30°, there is another protruded melt pool around 80–100 µm in height, exhibiting fused particles around itself, which explains the dross formation (Figure 3c,d). Figure 3d points out that a columnar grain structure is formed at the interface between the melt pool and particles in the powder bed. Following the protrusion of the melt pool into the powder bed, the contact between the melt pool boundary and the particles’ surface creates an interface, which alters the thermal gradient and affects the solidification dynamics. In other words, there should have been a relatively higher thermal gradient between the solid particle and liquid melt pool favoring a columnar grain structure. An increase in the fraction of fused particles onto the surface may end up with a relatively large columnar grain structure propagating from the material surface to the inside of the material up to 10–20 µm. It is well known that the columnar grain structure morphology may cause anisotropy and/or deterioration in the mechanical properties [54].

The Scanning Electron Microscopy (SEM) images obtained from the side view of CS-30°-125 µm are depicted in Figure 4a–c. As shown in Figure 2, the contact-free support and the overhang surface of the part are fused together during the printing of CS-30°-125 µm. Figure 4a exhibits the solidified large melt pools indexed with orange arrows at the overhang surface extending along the contact-free support surface. As shown below, those are the melt pools directly protruding into the powder bed. Provided that their height is larger than the predetermined gap distance, the support structure, and overhang surface fuse together. On the other hand, Figure 4b shows the depth of field view of the partially melted particles that are fused to the melt pool. According to Figure 4c, a particle could also contribute to the fusion of support structure and melt pool by creating bonding between them (indexed as bluearrow). As a result, even if the height of the melt pool is smaller than the gap distance, a particle could act as a bridge by being sintered to both overhang and contact-free support surfaces. Recalling that the samples could not be separated from the contact-free supports, it is concluded that the sticking issue took place at relatively low gap distances for 30° and 45° oblique angles of samples due to the reasons mentioned below. When the oblique angle was decreased to 15°, the height of the melt pool increased, and this increment also increased the critical gap distance to avoid fusion.

Figure 5 illustrates the surface texture maps derived from FV microscope images of US-45°, CS-45°-250 µm, CS-45°-300 µm, US-30°, CS-30°-250 µm, CS-30°-300 µm, US-25°, CS-25°-250 µm, and CS-25°-300 µm, respectively. It is useful to note that peaks are represented as yellow and red colors whereas valleys are blueish. The change of the trend in the surface texture maps with decreasing oblique angle is consistent with the surface roughness results listed in Table 3. It is quite clear that the fraction of higher peaks and deeper valleys is far more than at 30° oblique angles compared to those of 45° revealing that the deviation from the mean plane or surface roughness is also higher. As shown in Figure 5a–c, the peak and valley locations get slightly smoother with the increasing gap distance at the 45° oblique angle. However, considering the 30° oblique angle (Figure 5d–f), especially the fraction of claret red locations, which represent the severe peaks above 0.15 mm in height, diminishes indicating that the average roughness of the contact-free samples at 250 and 300 µm gap distances decrease due to the alleviation of the severely high peaks. Similarly, when the surface texture maps of the same overhang surfaces seen in Figure 5g–i are examined, the height of the peaks and depth of the valleys seen on US-25° remarkably decrease when contact-free structures come into play at CS-25°-250 µm, and CS-25°-300 µm. Those results are inconsistent with the average roughness results, which showed that the decrement in the surface roughness due to the integration of contact-free support gets greater at lower oblique angles.

High roughness existing at overhang surfaces of additively manufactured samples brings about deep and sharp-tip valleys, which may cause a line-of-sight problem during FV and may end up with a higher fraction of point losses [55]. So, in addition to a relative comparison of the surface roughness values obtained through FV, it is useful to examine the surface profile towards the cross-section under OM and to observe the surface profile more realistically. For this purpose, Figure 6a–f shows the surface profile images of overhang surfaces at different oblique angles (25°, 30°, and 45°) and gap distances (unsupported, 250 and 300 µm). It is evident that the peak heights and valley depths remarkably increase when the oblique angle is decreased. Different types of formations are observed in the cross-sectional OM images, which are considered to drive the surface roughness of the overhang surfaces: (i) Attached particles to the surface, (ii) balling phenomenon, and (iii) inverted mushroom features. The attached particles to the surface are indexed by yellow arrows and as shown, all the overhang surfaces exhibit those independent of the oblique angle and gap distance. The balling phenomenon is especially observed at the overhang surfaces having an oblique angle of 30° and 15° as indicated by green arrows in Figure 6d–i. It is well known that the balling effect occurs by the solidification of a spherical droplet, which was splashed out from the liquid melt pool due to surface tension [56]. Thus, it can be stated that the local instabilities in the melt pool came into play at the overhang surfaces at lower oblique angles and this resulted in the local breaking up of the melt track and solidification of spherical particles. The other feature contributing to the surface roughness of the overhang surfaces are inverted mushroom-like structures indexed by blue arrows in Figure 6a,c,h. It is assumed that when the viscosity of the melt pool increases due to cool down, the melt pool tends to expand towards the nearby powder bed based on the surface tension resulting in an inverted mushroom-like shape after solidification. Additionally, there are crack-like sharp and deep valleys (in other words, undercuts) indexed by red arrows, which may deteriorate the fatigue properties since those regions come up as the potential for micro-crack propagation zones [9]. The important point is that the related undercuts are sharper and deeper on the overhang surfaces at 25° and 30° oblique angles compared to those of 45°. It is an expected phenomenon since the surface irregularities are more severe at lower angles and thereby, overhang surfaces relative at oblique angles are more prone to the sharp undercuts.

Figure 7a–f shows the top-view SEM images provided from the overhang surfaces of the samples. For the unsupported samples of US-45°, US-30° and US-25° shown in Figure 7a–c, the deterioration of the surface roughness is clear with decreasing oblique angle. It is observed that the particles fused to the surfaces (indexed as P) do not govern the surface roughness difference between the unsupported overhang surfaces since their fraction does not change remarkably at different oblique angles. However, overhang surface of US-30° exhibits protruding melt pools towards the powder bed compared to that of US-45°. Besides, when the oblique angle is decreased to 25°, the overhang surface of US-25° contains deep void regions (indexed as V) in addition to the bumpy melt pools observed at those of US-30°. Therefore, it could be said that bumpy melt pool topography and voids cause higher peaks and deeper valleys and thereby a more severe surface roughness at relatively lower oblique angles. In addition to the low oblique angles, gap distances of 250 and 300 µm provided an effective thermal dissipation, where the melt pool cooled down quickly and this may have resulted in a less bumpy melt pool topography and thereby, lower surface roughness compared to those of unsupported ones. On the other hand, the SEM images of contact-free supported overhang surfaces (see Figure 7d–f) do not show a remarkable difference in the fused particle concentration similar to those of unsupported ones. In addition, the SEM image of the surface topography does not show a significant difference between the supported and unsupported ones at 30° and 45° oblique angles. However, when the overhang surfaces of US-25° and CS-25°-300 µm are compared (see Figure 7c,f), a dramatic change in the surface topography appears. As mentioned above, while the overhang surface of US-25° contains large and deep voids, the voids disappear at CS-25°-300 µm. This is consistent with the surface texture maps shown in Figure 5a–c, where the blue regions representing deep valleys larger than 150 µm remarkably decrease from that of US-25° to CS-25°-300 µm. It is also important to note that the balling effect has also been observed especially at US-25° and CS-25°-300 µm. Regarding Figure 7a–c again, the claret red regions representing the high peaks larger than 150 µm are assumed to belong to the regions where the balling effect took place and their fraction also decreases with the implementation of contact-free structures. Since the heat dissipation is increased by the implementation of the support structures, the balling effect might have been alleviated during printing compared to the unsupported samples.

Figure 8 shows the surface roughness parameters (Ra, Rq, and Rz) vs. oblique angle plots of the overhang surfaces at different oblique angles (25°, 30° and 45°) and gap distances (unsupported, 250 and 300 µm). All surface roughness parameters follow a similar trend as a function of oblique angle regardless of the gap distance values. Regarding the 45° oblique angle, Ra, Rq, and Rz are closer to each other at all gap distances, which shows that the implementation of a contact-free support structure beneath the overhang surfaces does not make a significant contribution to enhancing surface topography at 45° oblique angle. However, the contribution of contact-free support structures becomes more apparent as the angle is decreased. Regarding the 25° oblique angle, Ra, Rq, and Rz are dramatically lower at CS-25°-250 µm and CS-25°-300 µm samples compared to those of US-25°. As a result, the surface roughness values efficiently decrease with the implementation of the contact-free support structures, especially at relatively lower oblique angles (25° and 30°) and relatively higher gap distances (250 and 300 µm). So, while designing a contact-free supported part, a “critical gap distance” should be considered, which does not result in binding by providing an effective heat dissipation. In this way, the surface quality of the overhang surfaces could be enhanced at low oblique angles. As mentioned previously, the surface roughness characteristics of the samples directly affect the fatigue properties. Therefore, it is believed that those results shed light on the potential of contact-free support structure implementation on the overhangs, where the surface removal processes are not applicable due to some surface treatment limitations in complex geometries.

## 4. Conclusions

During this study, samples having overhang surfaces with different oblique angles of 15°, 25°, 30°, and 45° were supported with contact-free support structures separated from the overhang surface with gap distances of 100, 125, 150, 250, and 300 µm were produced using L-PBF and a Co-Cr-Mo alloy. Moreover, unsupported specimens were also manufactured at the same overhang angles for comparison. Their surface quality was evaluated in terms of roughness (Ra, Rq, and Rz) as well as OM and SEM characterizations. According to the results of this study, the following conclusions can be drawn:The gap distance for contact-free support structures is a very critical factor and needs to be optimized. The fusion of the contact-free supports and overhang surfaces mainly occurs at lower gap distances, namely 100, 125, and 150 µm which are 2, 2.5, and 3 folds of the selected layer thickness. The fusion is mainly due to the melt pool depth exceeding 2–3 layer thicknesses. At the lowest oblique angle of 15°, all the produced specimens exhibited either fusion with the contact-free support or a very poor surface quality impossible to measure.The OM and SEM micrographs showed that protrusion of the melt pool into the powder bed results in direct contact of the melt pool with the support surfaces and/or acting of particles in the powder bed as bridges connecting the melt pool and support structure at lower gap distances.The surface roughness values (Ra, Rq, and Rz) of the CS-25°-250 µm, CS-25°-300 µm, CS-30°-250 µm, and CS-30°-300 µm significantly decrease compared to their unsupported counterparts. Thus, the decrement rate in surface roughness parameters is more remarkable at higher gap distances. Ra, Rq, and Rz decrease by around 2 times in CS-25°-300 µm compared to those of unsupported cases.The OM and SEM micrographs also showed that the surface texture is determined by several factors such as attached particles, inverted mushroom-like features, balling phenomena, undercuts, bumpiness of the melt pool, etc.

## Figures and Tables

**Figure 1 materials-15-03765-f001:**
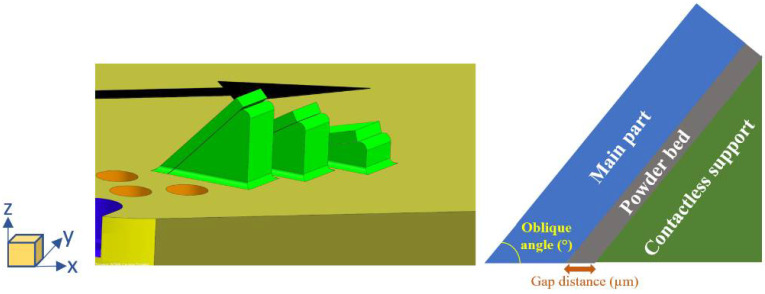
Representative sample design.

**Figure 2 materials-15-03765-f002:**
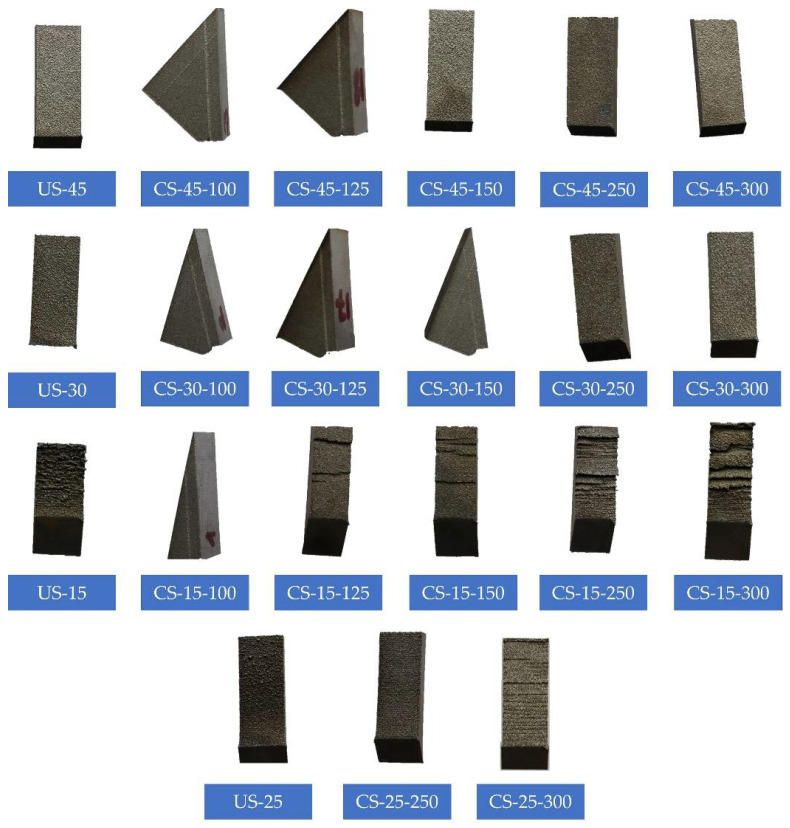
Contact-free supported geometries printed via DMLM (Coupons like pyramids are the ones that fused to support. Remaining ones represent overhang areas of corresponding coupons).

**Figure 3 materials-15-03765-f003:**
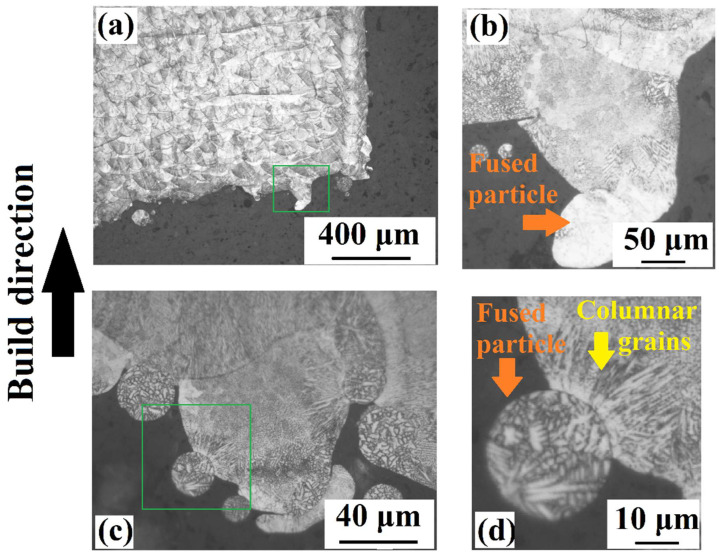
The cross-sectional OM images obtained from: (**a**) US-30° (×50) and (**b**) US-30° (×200), (**c**) US-45° (×1000) and (**d**) US-45° (×2000).

**Figure 4 materials-15-03765-f004:**
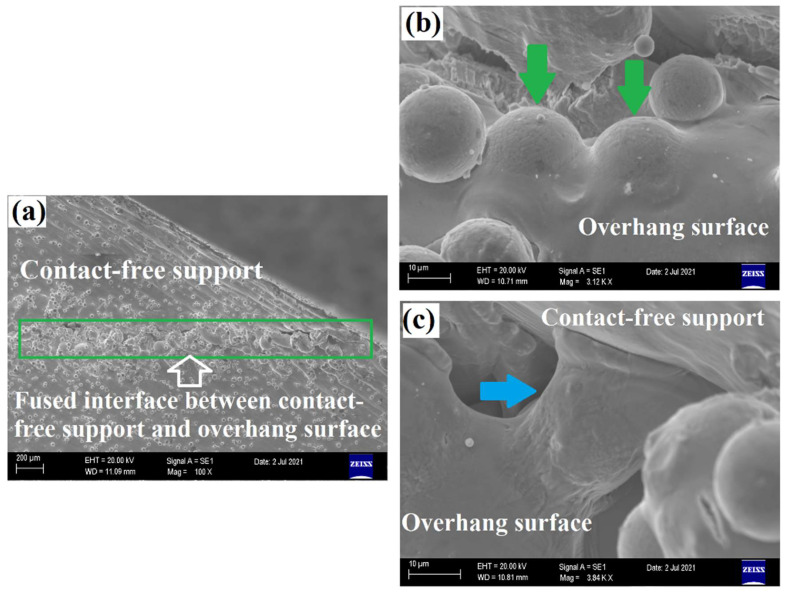
SEM images obtained from the side view of CS-30°-125 µm: (**a**) Macro view of region of fusion, (**b**) Partially melted particles on the melt pool surface, and (**c**) A particle created a bonding between overhang surface and contact-free support structure.

**Figure 5 materials-15-03765-f005:**
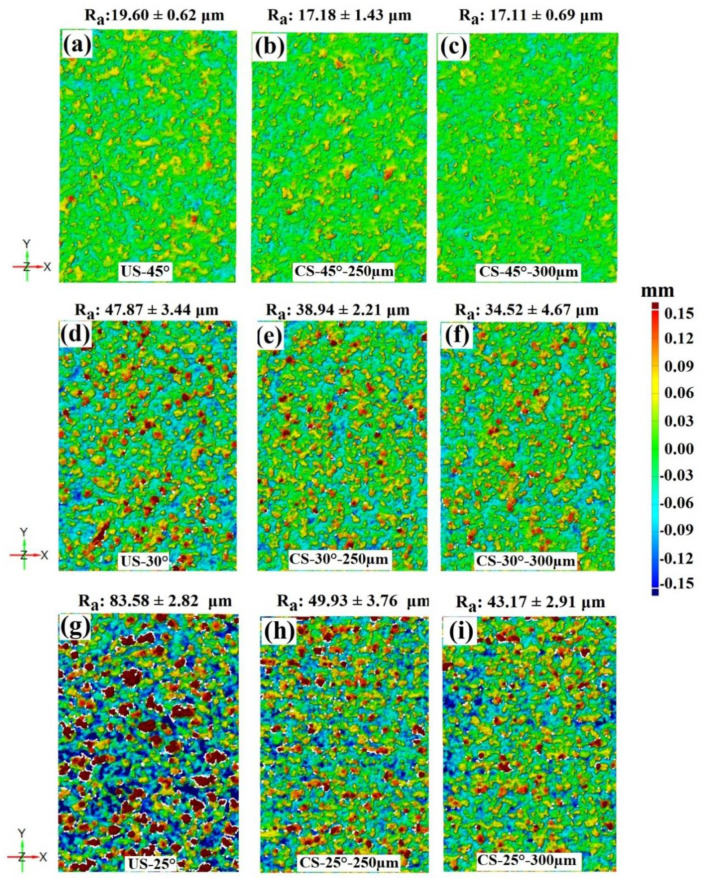
Surface texture maps derived from FV microscope images: (**a**) US-45°, (**b**) CS-45°-250 µm, (**c**) CS-45°-300 µm, (**d**) US-30°, (**e**) CS-30°-250 µm, (**f**) CS-30°-300 µm, (**g**) US-25°, (**h**) CS-25°-250 µm, and (**i**) CS-25°-300 µm.

**Figure 6 materials-15-03765-f006:**
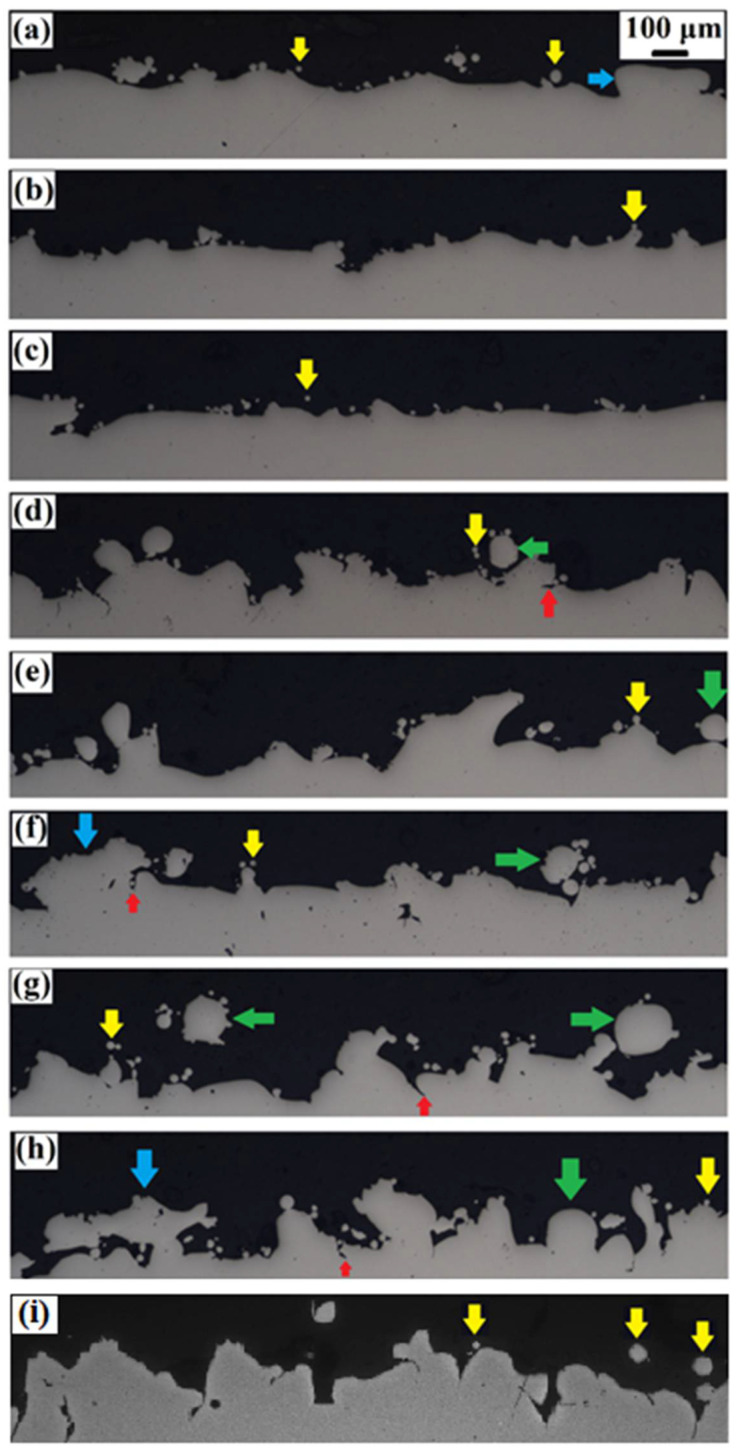
Cross-section OM images obtained from: (**a**) US-45°, (**b**) CS-45°-250 µm, (**c**) CS-45°-300 µm, (**d**) US-30°, (**e**) CS-30°-250 µm, (**f**) CS-30°-300 µm, (**g**) US-25°, (**h**) CS-25°-250 µm, and (**i**) CS-25°-300 µm.

**Figure 7 materials-15-03765-f007:**
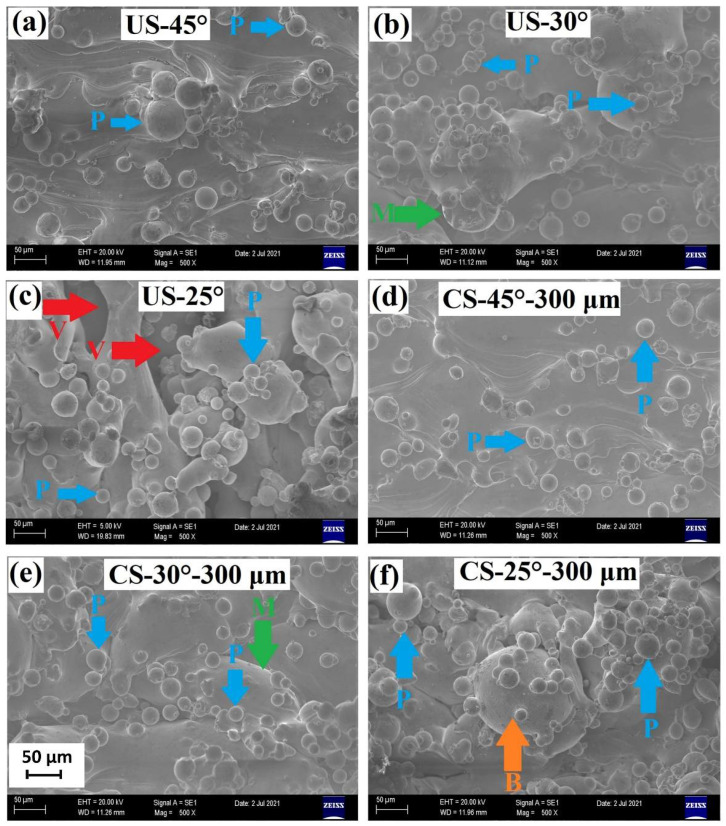
SEM images taken from the overhang surfaces of the samples (z-projection): (**a**) US-45°, (**b**) US-30°, (**c**) US-35°, (**d**) CS-45°-300 µm, (**e**) CS-30°-300 µm and, (**f**) CS-25°-300 µm.

**Figure 8 materials-15-03765-f008:**
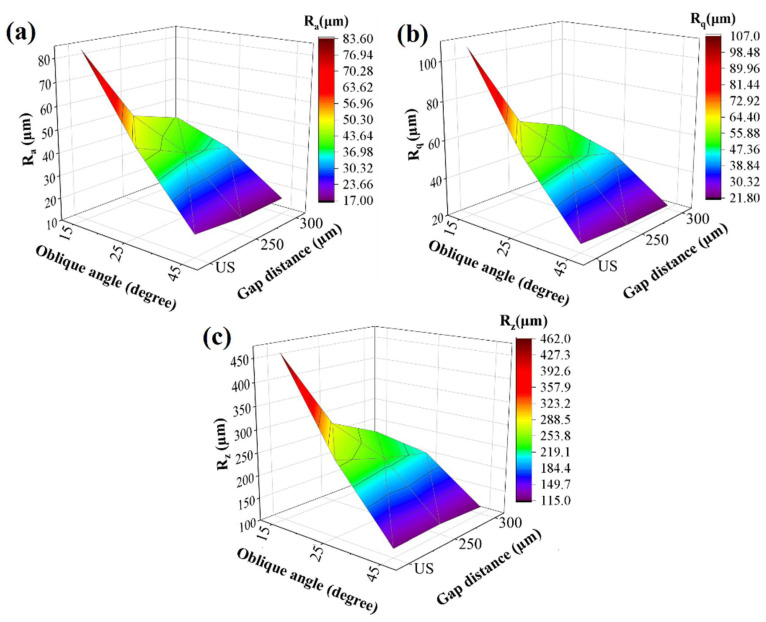
3D plots of surface roughness parameters: (**a**) Ra vs. oblique angle vs. gap distance, (**b**) Rq vs. oblique angle vs. gap distance, and (**c**) Rz vs. oblique angle vs. gap distance.

**Table 1 materials-15-03765-t001:** Chemical composition of Co-Cr-Mo powder.

Element	Wt. %
Chromium	27.68
Molybdenum	5.82
Silicon	0.48
Iron	0.36
Manganese	0.25
Nickel	0.23
Nitrogen	0.11
Carbon	0.10
Tungsten	0.04
Oxygen	0.02
Phosphorus	<0.005
Sulfur	0.004
Boron	0.001
Cobalt	Balance

**Table 2 materials-15-03765-t002:** Test plan.

Sample #	Support Type	Oblique Angle	Gap Distance	Sample ID
1	Unsupported	45°	-	US-45°
2	Contactless Support	45°	100 µm	CS-45°-100 µm
3	Contactless Support	45°	125 µm	CS-45°-125 µm
4	Contactless Support	45°	150 µm	CS-45°-150 µm
5	Contactless Support	45°	250 µm	CS-45°-250 µm
6	Contactless Support	45°	300 µm	CS-45°-300 µm
7	Unsupported	30°	-	US-30°
8	Contactless Support	30°	100 µm	CS-30°-100 µm
9	Contactless Support	30°	125 µm	CS-30°-125 µm
10	Contactless Support	30°	150 µm	CS-30°-150 µm
11	Contactless Support	30°	250 µm	CS-30°-250 µm
12	Contactless Support	30°	300 µm	CS-30°-300 µm
13	Unsupported	15°	-	US-15°
14	Contactless Support	15°	100 µm	CS-15°-100 µm
15	Contactless Support	15°	125 µm	CS-15°-125 µm
16	Contactless Support	15°	150 µm	CS-15°-150 µm
17	Contactless Support	15°	250 µm	CS-15°-250 µm
18	Contactless Support	15°	300 µm	CS-15°-300 µm
19	Unsupported	25°	-	US-25°
20	Contactless Support	25°	250 µm	CS-25°-250 µm
21	Contactless Support	25°	300 µm	CS-25°-300 µm

**Table 3 materials-15-03765-t003:** Ra, Rq, and Rz values obtained from the overhang surfaces of different oblique angles (15°, 30° and 45°) and gap distances (unsupported, 100, 125, 150, 250, and 300 µm).

Sample ID	Support Type	Oblique Angle	Gap Distance
US-45°	19.60 ± 0.62	24.49 ± 0.45	119.40 ± 4.04
CS-45°-100 µm	Fused to the support	Fused to the support	Fused to the support
CS-45°-125 µm	Fused to the support	Fused to the support	Fused to the support
CS-45°-150 µm	20.22 ± 0.54	26.69 ± 0.23	141.29 ± 5.22
CS-45°-250 µm	17.18 ± 1.43	22.35 ± 2.46	122.32 ± 10.23
CS-45°-300 µm	17.11 ± 0.69	21.81 ± 0.80	115.68 ± 2.94
US-30°	47.87 ± 3.44	58.90 ± 4.27	264.46 ± 24.62
CS-30°-100 µm	Fused to the support	Fused to the support	Fused to the support
CS-30°-125 µm	Fused to the support	Fused to the support	Fused to the support
CS-30°-150 µm	Fused to the support	Fused to the support	Fused to the support
CS-30°-250 µm	38.94 ± 2.21	49.77 ± 3.65	232.41 ± 9.40
CS-30°-300 µm	34.52 ± 4.67	43.25 ± 5.81	208.89 ± 22.50
US-15°	Poor surface quality	Poor surface quality	Poor surface quality
CS-15°-100 µm	Fused to the support	Fused to the support	Fused to the support
CS-15°-125 µm	Poor surface quality	Poor surface quality	Poor surface quality
CS-15°-150 µm	Poor surface quality	Poor surface quality	Poor surface quality
CS-15°-250 µm	Poor surface quality	Poor surface quality	Poor surface quality
CS-15°-300 µm	Poor surface quality	Poor surface quality	Poor surface quality
US-25°	83.58 ± 2.82	106.96 ± 5.03	461.50 ± 3.07
CS-25°-250 µm	49.93 ± 3.76	62.19 ± 4.70	282.50 ± 18.29
CS-25°-300 µm	43.17 ± 2.91	52.28 ± 3.38	231.52 ± 11.34

## Data Availability

Not applicable.

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
