# Peer review of "Contact-Free Support Structures for the Direct Metal Laser Melting Process"

_materials, 2022, doi:10.3390/ma15113765_

Round 1

Reviewer 1 Report

  • Abstract: could be more concise
  • Introduction
    • Shorter first paragraph would enhance readability
    • Authors should reflect on the typical understanding of the journal's readership e.g. a materials science and not necessarily someone with an AM background. With this in mind they may wish to further explain some of the processes / fundamentals of direct laser melting 
    • The authors could substantiate more statements, e.g. add typical Ra when discussing the reference by Cooper et al.
    • The authors should justify why they have performed their study with Co-Cr-Mo powder
  • Materials and Methods
    • Table 1 could be supplementary
    • The authors should had the printing parameters used, e.g. laser power, speed etc. 
  • Results
    • Figure 2 should add a scale bar
    • Why are the images in figure 2 in different orientations?

Author Response

Dear Reviewer,

You can find the comments in the attached file.

Regards

Reviewer 2 Report

The aim of this study is to understand the effect of different parameters, such as gap distance and inclination angle, on the topography and microstructural properties of the overhang surfaces. The final results indicated that contact-free supports have a positive effect on decreasing surface roughness at all build angles when the gap distance is correctly set to avoid sintering of the powder in between the overhang and supports or to avoid too large gaps eliminating the desired effect of the higher thermal conductivity. The authors need to address the following issues/comments for publications by the journal:

  1. The abstract lacks stating the gap that is not filled using the available literature and this research is going to fill. Please briefly describe the unsolved problem that you've solved with the presented research.
  2. In introduction, please explain why the CoCrMo alloy is chose as the test materials. Are other additive materials also suitable for the conclusions in this paper?
  3. Page 11, Fig.6: The letter numbers are too big but the scale bars are too small to read.
  4. Page 12, Fig.7: The scale bars are too small to read.
  5. In this paper, only the surface roughness is used to evaluate the AM forming quality. Please add the comparative test of microstructure or mechanical properties to comprehensively evaluate the influence of different forming conditions.

Author Response

(The authors gave the same response as above.)
